# Biological Variation and Reference Change Value of Routine Hematology Measurands in a Population of Managed Bottlenose Dolphins (*Tursiops truncatus*)

**DOI:** 10.3390/ani13081313

**Published:** 2023-04-11

**Authors:** Federico Bonsembiante, Carlo Guglielmini, Michele Berlanda, Pietro Fiocco, Barbara Biancani, Carla Genovese, Silvia Bedin, Maria Elena Gelain

**Affiliations:** 1Department of Animal Medicine, Production and Health, University of Padua, 35020 Legnaro, PD, Italy; 2Department of Comparative Biomedicine and Food Science, University of Padua, 35020 Legnaro, PD, Italy; 3Zoomarine Italia, 00071 Torvaianica, RM, Italy

**Keywords:** within-individual variability, between-individuals variability, index of individuality, marine mammals, complete blood cell count

## Abstract

**Simple Summary:**

Calculating hematological reference intervals for bottlenose dolphins poses a challenge due to a limited number of reference individuals. However, individual reference intervals (iRIs) can be used to overcome this limitation. We evaluated the biological variations in hematological measurands and calculated the index of individuality (IoI) and the reference change value (RCV), which enable the production of iRIs, in healthy managed bottlenose dolphins. Analysis of IoI indicates that the use of iRIs is suitable for most hematological measurands. Furthermore, the calculated RCV can be applied to other dolphins that have undergone serial hematological exams. These tools provide valuable information for interpreting hematologic exams in managed bottlenose dolphins.

**Abstract:**

Hematological analyses are particularly useful in assessing a dolphin’s health status. However, the creation of appropriate reference intervals for this species is difficult due to the low number of reference individuals. The implementation of individual reference intervals (iRIs) allows researchers to overcome this limitation and, moreover, also consider the within-individual variability. The aims of this study were (1) to evaluate the biological variations in some hematological measurands, including erythrocytes (RBC), hematocrit (Hct), mean cellular volume and hemoglobin content (MCV and MCHC, respectively), RBC distribution width (RDW), leukocytes (WBC), and platelets (PLT); and (2) to calculate the index of individuality (IoI) and reference change value (RCV), which enable the production of iRIs, in healthy managed bottlenose dolphins. Seven dolphins were included, and the results of six hematological exams were analyzed for each animal. Analytical imprecision (CV_a_), within-dolphin variation (CV_i_), and between-dolphins variations (CV_g_) were calculated, and the IoI and RCV were derived for each measurand. All the hematological measurands had intermediate IoI except WBC, for which Iol was low. The calculated RCV ranged from 10.33% (MCV) to 186.51% (WBC). The results reveal that the majority of hematological measurands have an intermediate level of individuality in dolphins, and thus the application of iRIs is appropriate. The calculated RCV can also be applied to other managed dolphins and could be useful in interpreting serial CBC exams.

## 1. Introduction

Complete blood cell count (CBC) is a mainstay of veterinary diagnostics and provides fundamental information to assess the health status of an individual. The interpretation of a CBC exam is mainly performed by comparing the patient’s results to reference intervals (RIs). The latter are usually population-based (pRIs) and comprise the central 95% of a healthy reference population [1]. The optimal number of reference individuals needed to calculate the pRIs is 120, while the pRIs cannot be calculated if the reference population is less than 40 individuals. If there are between 41 and 119 reference individuals, pRIs can be calculated, but there is an increase in uncertainty for the determination of pRIs with a decrease in reference individuals [1]. When working with exotic or non-domestic species, it is often impossible for a single laboratory to receive samples from enough reference individuals to create its own pRIs, and it is necessary to adopt multicenter RIs [2]. However, this requires that all the laboratories involved adopt strict and common procedures (e.g., same control material) to meet the same quality control goals [1]. Another shortcoming in the calculation of pRIs is the lack of within-individual variability (CV_i_), which represents the physiological fluctuation of the values of a determined analyte around the homeostatic set point (HSP) of an individual [3].

The use of individual-based RIs (iRIs) allows one to overcome these pRI flaws for the following reasons: They are calculated one by one from serial measurements from each healthy individual, they take into account the CV_i_ and analytical imprecision (CV_a_), which could affect clinical laboratory results [4], and they require fewer reference individuals to be calculated [5]. Furthermore, recent studies on biological variation have demonstrated that several hematological and biochemical measurands, independent from the species, are characterized by high individuality. For this reason, they are better interpreted using the iRIs [6]. The individuality of an analyte (i.e., the index of individuality [IoI]) represents the relationship between CV_g_ (i.e., the variation between individuals) and CV_i_: analytes with high IoI have higher CV_g_ than CV_i_ and, as a consequence, wide pRIs that are not sensitive enough to detect a significant change in a test result. On the contrary, analytes with low IoI have lower CV_g_ than CV_i_ and narrow pRIs [7]. The pivotal point in the generation of iRIs is the calculation of the reference change value (RCV), defined as ‘‘that difference between 2 consecutive test results in an individual that is statistically significant in a given proportion of all similar persons’’ [8]. The calculation of RCV is based on the CV_i_ and the CV_a_ and allows the clinician to understand whether the percentage of changes observed between HSP and a new test result is clinically relevant or is attributable to physiological fluctuations over time [7]. Once the RCV is calculated, the iRIs for each individual are calculated by adding and subtracting the RCV to and from the mean value of a determinate analyte.

Bottlenose dolphins (*Tursiops truncatus*) are one of the most common marine mammal species housed in aquaria worldwide, and they undergo periodic clinical assessment to evaluate their health status according to specific legislation and preventive medicine programs [9]. The clinical evaluation of marine mammals is challenging because the typical clinical signs of disease found in humans and domestic animals are difficult to recognize or interpret in wild/exotic species. Therefore, a good preventive medicine program is fundamental, and laboratory analyses and interpretation of their results according to appropriate RIs with the best diagnostic accuracy have considerable importance in disease identification [9]. Reference intervals for CBC values in bottlenose dolphins have been reported in the literature, including results from free-ranging and captive dolphins [10,11,12]. The conditions of populations under human care can vary according to different life conditions (e.g., controlled water environment versus open ocean access, pathogen exposure, feeding, and stressful situations) [11,12]. Thus, results reported in the literature should be evaluated accordingly.

Most studies evaluating the biological variation were focused on domestic animal species such as dogs [13,14], cats [15,16], horses [17], and cows [18]. Furthermore, biological variation has also been evaluated in some non-domestic species, both mammals and non-mammals, due to the small number of reference individuals needed for the calculation [19,20,21,22,23]. To date, no studies have evaluated the biological variation in hematological measurands in dolphins. Thus, the aims of this study were (1) to determine the biological variations and (2) to calculate the IoI and RCV of hematological measurands in a managed population of bottlenose dolphins.

## 2. Materials and Methods

Samples obtained from seven clinically healthy bottlenose dolphins (*Tursiops truncatus*) housed in an Italian zoological park (Zoomarine Italia, Torvaianica, Rome, Italy) were used for the present study. Four females and three males, ranging in age from 6 to 40 years (Table 1), were monitored for a period of 3 years, and for each animal, the results of 6 CBC exams were included (two CBC exams/year/animal). The animals were considered healthy based on medical history and regular physical examination performed as part of the preventive medicine program. The dolphins were housed and handled in accordance with the Italian Zoo Directive Law (DL 73/2005), and blood samples were obtained according to the D.M. 469/2001, which establishes the management objectives and prescriptions to maintain the species *Tursiops truncatus* under human care. All dolphins maintained in the facility were trained to participate voluntarily in veterinary and husbandry procedures in order to guarantee regular routine diagnostic analysis. In particular, the animals were trained for voluntary venipuncture, presenting the fluke for blood collection without physical restraint. Around two milliliters of blood were collected in K3-EDTA tubes (S-Monovette, Sarstedt AG & Co. KG, Nümbrecht, Germany), and, for each sample, a blood smear was performed immediately after sampling. K3-EDTA blood samples and blood smears were shipped refrigerated to the Clinical Pathology Laboratory of the Veterinary Teaching Hospital of the University of Padova (VTH-UP). Hematological exams were performed within 24 h from collection using an automated hematologic analyzer (ADVIA 120, Siemens Healthcare Diagnostic Inc., Deerfield, IL, USA) equipped with multi-species software designed for use in veterinary medicine. We employed the human setting of the analyzer because of the similarity of the hematocrit (Hct) and mean cellular volume (MCV) between the two species. The leukocyte differential count was performed manually by counting five times 100 leukocytes, and the average percentages allow one to calculate the absolute number of each leukocyte subpopulation. Red blood cell indices, including number of erythrocytes (RBC), Hct, hemoglobin (Hgb), MCV, mean cellular hemoglobin content (MCHC), red blood cell distribution width (RDW), number of leukocytes (WBC) with their manual differential count, and number of platelets (PLT) were evaluated. All blood samples were collected for clinical diagnostic purposes, and no ethical approval was needed.

Quality control procedures included daily testing of the hematology analyzer with a manufacturer-supplied control (Siemens 3 in 1 TESTpoint Hematology Control, Siemens Healthcare Diagnostic Inc., Deerfield, IL, USA). Furthermore, the Clinical Pathology Laboratory of the VTH-UP participated in the Randox International Quality Assessment Scheme (RIQAS) Hematology External Quality Assessment.

### Data Analysis

Data were analyzed using commercially available statistical software (MedCalc Statistical Software, version 19.2.6, 2020, Mariakerke, Belgium) and Microsoft^®^ Excel^®^ 2019. The Tukey test was used to detect the presence of outliers, while data distribution was assessed by histogram visual inspection and use of the Shapiro-Wilks test. The non-normally distributed parameters were log transformed. Analytical (CV_a_), between-dolphins (CV_g_), and within-dolphin (CV_i_) variations were calculated using the formula $\frac{Standard\;deviation}{mean\;analyte\;value}*100$ (for normally distributed analytes) and $\sqrt{e^{\mathrm{VAR}\left(LNx\right)}-1}$ * 100 (for log-transformed analyte), as proposed by Fokkema and colleagues [25] (Table 2). The IoI was calculated using the formula $\;\mathrm{CVg}/\sqrt{{\mathrm{CVi}}^{2}+{\mathrm{CVa}}^{2}}$ [7]. Based on this formula, analytes with high individuality were defined when IoI was >1.67; analytes with low individuality were defined when IoI was <0.7. When IoI was between 0.7 and 1.67, analytes were defined as having an intermediate individuality [5,6].

The formula to calculate the RCV was $Z*\sqrt{2*}\sqrt{{\mathrm{CVi}}^{2}+{\mathrm{CVa}}^{2}}$ for normally distributed analytes. The RCV formula for analytes that required logarithmic transformation was described by Fokkema and colleagues [25] and used the log-transformed variance component (σ) to calculate non-symmetric RCVs: $+Z*\sqrt{[2*(\sigma_{i}^{2}+\sigma_{a}^{2})]}$ for increasing values and $-Z*\sqrt{[2*(\sigma_{i}^{2}+\sigma_{a}^{2})]}$ for decreasing values (Table 2). The Z value was set at 1.96 since it was important to interpret both an increase and a decrease in the results [15].

## 3. Results

In total, we analyzed 504 values (12 measurands [RBC, Hct, Hgb, MCV, RDW, PLT, WBC, % of neutrophils, lymphocytes, monocytes, eosinophils, basophils] for 7 animals evaluated 6 times). Among these, there were 16 suspected outliers, but these were not eliminated. Erythrocytes, Hct, Hgb, MCV, RDW, and PLT were normally distributed, while WBC required log transformation. The CV_i_, CV_g_, CV_a_, CV_a_/CV_i_, IoI, and RCV values are reported in Table 3. Desirable imprecision (CV_a_/CV_i_ < 0.5) was achieved for five analytes (RBC, Hct, Hgb, PLT, and WBC), while for others (MCV, MCHC, and RDW), the CV_a_/CV_i_ ratios were between 0.57 and 0.66. Imprecision was not acceptable for the leukocyte subpopulations obtained by manual differential count (CV_a_/CV_i_ ratio ranging from 0.73 to 1.41). All analytes except WBC (IoI = 0.66) had an intermediate IoI, ranging from 0.78 to 1.26. Calculated RCVs ranged from 10.33% (MCV) to 17.40% (Hct) for normally distributed data, while for WBC, they ranged from 53.62% to 186.51% when evaluating a decreased or increased WBC number, respectively.

Table 4 reports the comparison of CV_i_ of bottlenose dolphins with the CV_i_ of other mammals’ species. Figure 1 reports the iRIs of the seven bottlenose dolphins included in the present study compared to the pRIs obtained from the literature [2,10].

## 4. Discussion

To our knowledge, this is the first study to evaluate the biological variation of CBC parameters in a group of managed dolphins. The first issue to overcome was the appropriate setting selection in the automated hematologic analyzer. ADVIA120 is a flow cytometric-based hematological analyzer equipped with a multi-species software to correctly analyze blood samples from the most common domestic mammal species. Unfortunately, there is no dedicated setting for dolphins. The common approach to evaluating blood samples from a species without a dedicated setting is to use the available setting for the species with the closest value of Hct [27,28]. Since in ADVIA120 the Hct is calculated from the MCV and the RBC and not directly measured, we used the human setting due to the similarity of MCV and Hct in the two species. We obtained an acceptable (CV_a_/CV_i_ < 0.5) or minimal (CV_a_/CV_i_ < 0.75) imprecision for all analytes measured by the automatic hematologic analyzer. These degrees of imprecision mean that an analytical imprecision influence of less than 12% or 25%, respectively, of the total variability of a sample and a CV_a_/CV_i_ threshold < 0.75 is considered the minimum acceptable imprecision [3,29]. The low imprecision reported in our study was expected, as the CV_a_ of all the automatic measured analytes was below the maximum allowable total error (TE_a_) threshold proposed in the ASVCP guidelines [30]. The TE_a_ for RDW-CV was not included in the ASVCP guidelines; however, RDW is calculated from the MCV. Thus, its CV_a_ is related to the imprecision of the mean cell volume. In our study, the CV_a_ was 1.85% and 3.20% for MCV and RDW-CV, respectively, and both values can be considered to be acceptable imprecision values. Furthermore, the VTH-UP Clinical Pathology Laboratory participates in the RIQAS external quality control program, and the analytical performances of the hematologic analyzer have always been considered acceptable. On the contrary, imprecision was not acceptable (CV_a_/CV_i_ > 0.75) for the leukocyte subpopulations due to the high CV_a_, which ranged from 10.16% (neutrophils) to 41.93% (monocytes). In our study, the CV_a_ for the leukocytes was calculated on differential counts performed 5 times, considering 100 leukocytes each, since we considered the differential counts provided by the hematologic analyzer to be unreliable. ADVIA120 differentiates leukocyte subpopulations according to their myeloperoxidase content and size [31]. Since the myeloperoxidase content could be different between leukocytes from different species, and thus the automatic classification could be erroneous, we preferred to perform a manual differential count to obtain the percentages of the different leukocyte subpopulations. This approach has also been reported for other non-domestic species [27,32]. Similar CV_a_s for manual differential counts were obtained in dogs [33] and horses [34]. These findings are likely attributable to the low number of leukocytes counted and the evaluation of different microscopic fields during the five readings [35]. A possible solution to improve CV_a_ is to include a higher number of leukocytes during differential counts, but this is time-consuming and not applicable in routine workflow. Due to the inacceptable imprecision, we did not calculate the RCV for the leukocyte subpopulations.

The results demonstrated that all of the measurands examined, except WBC, had intermediate IoI. Similar results regarding the IoI of hematological measurands were obtained for cows [18] and for laboratory beagles [13]. The animals of these studies lived in the same environment and received the same food, similar to the dolphins in the present study, leading to a low CV_g_. Campora and colleagues (2018) suggested that the best approach for intermediate IoI is to interpret test results according to both pRIs and iRIs [6]. Moreover, in the same article, it is reported that, in the case of serial measurements, it is appropriate to analyze the RCV between two consecutive measurements rather than compare test results to pRIs [6]. In our laboratory, the pRIs for dolphins were not calculated due to the low number of reference individuals; therefore, we decided to compare new test results to the RCV and iRIs. All dolphins included in the present study are subject to periodic CBC exams, and we propose to interpret changes in the WBC number between two consecutive CBC exams in the same animal by evaluating the RCV. If the percentage of variation is higher than the RCV, the change should be considered abnormal and cannot be justified by random fluctuation. The calculated two-sided RCV varied between 10.33% for MCV and 186.51% for WBC. We calculate the two-sided RCV instead of the one-sided RCV because it is important to appreciate both decreases and increases in the hematological parameters. The RCV is the % of variation between the HSP and a new test result or between two consecutive test results [36]. This means that in MCV (RCV = 10.33%), for example, a difference between the HSP and a new test result lower than 10.33% could be attributable to biological variation, while if it exceeds 10.33%, it can be considered abnormal. When evaluating the difference between HSP and a new test result or between two consecutive test results, it is important to minimize the variation due to preanalytical errors, since this is the main source of error in veterinary laboratory, ranging from 52% to 77% of total errors [37]. In our study, the preanalytical variations were low since all the procedures were highly standardized and only samples collected in less than 24 h and without hemolysis were included.

In the literature some pRIs for bottlenose dolphins under human care are available, and if they are compared with the IRIs of our animals, it is noticeable that some parameters in our samples would have been considered differently. For example, HCT would be considered out of range in three dolphins using the pRIs of Gulland and colleagues [10] and within the reference range using the pRIs of Lauderdale and colleagues [2]. The same observation is possible for MCV, MCHC, and PLT.

The retrospective nature of this study leads to some limitations. First, blood samples were collected and analyzed for diagnostic purposes, and thus duplicate measurements of the same sample were not performed. This hampered the use of other statistical tests suggested by Freeman and colleagues [5], and we used the classical formula $\frac{Standard\;deviation}{mean\;analyte\;value}*100\;$ to calculate CV_i_ and CV_g_. Furthermore, CV_a_ was determined using control material rather than the dolphins’ blood samples. However, the use of control material to assess CV_a_ is considered reasonable [7]. Second, a long time passed (about three years) between the collection of the first and last samples. Age has been reported to influence WBC, with an increase in leukocytes in older cetaceans [11]. This agrees with our results, where the only geriatric animal (animal F) has the highest iRI for WBC (Figure 1). However, dolphins are long-lived animals, and they could live more than 60 years [38]. In the three-year study period, there were no changes in age class for any dolphin. Therefore, it was unlikely that possible age-associated changes in hematologic measurands would be significant enough to alter the results of our study. Third, our population was made up of seven dolphins, a lower number than the 10–15 animals suggested [5]. However, when working with non-domestic animals, it is difficult to have a larger population, and we decided to include only animals that had at least six serial CBC results and that were considered undoubtedly healthy. Finally, due to the low number of animals, it was not possible to divide the animals according to their gender or age.

## 5. Conclusions

In this study, we evaluated, for the first time, the biological variation in a population of managed bottlenose dolphins. The results reveal that the majority of hematological measurands in dolphins are characterized by an intermediate individuality and that the application of iRIs is appropriate. The calculated RCV can also be applied to other populations of managed bottlenose dolphins and could be useful in interpreting their serial CBC exams. Further studies that include a higher number of animals are warranted.

## Figures and Tables

**Figure 1 animals-13-01313-f001:**
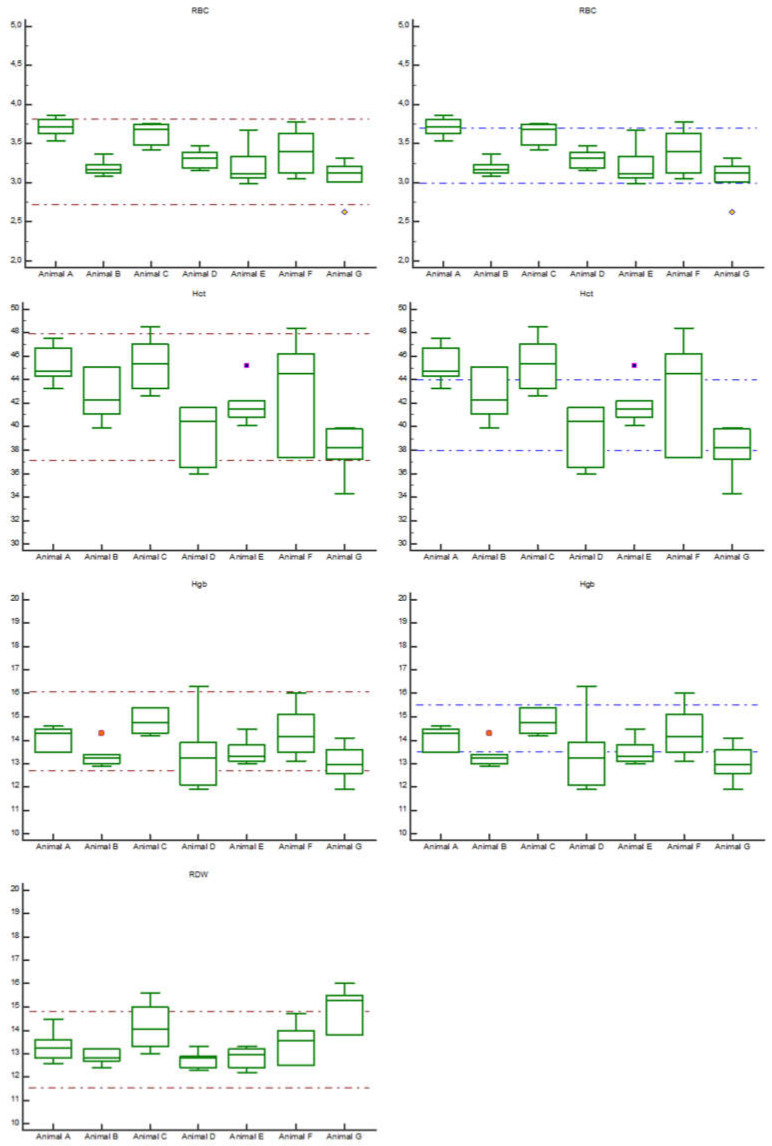
Comparison of individual reference intervals (iRIs) with population-based reference intervals (pRIs) obtained from the literature. The central box represents the values from the lower to upper quartiles. The middle line represents the median. A line extends from the minimum to the maximum value, excluding suspected outliers, which are displayed as separate points. The dotted lines represent the population-based reference intervals (pRIs) for managed bottlenose dolphins published by Lauderdale and colleagues [2] (in red) and by Gulland and colleagues [10] (in blue). The pRI for red blood cell distribution width (RDW) was not calculated by Gulland and colleagues. RBC = number of erythrocytes (10^9^/µL), Hct = hematocrit (%), Hgb = hemoglobin (g/dL), MCV = mean cellular volume (fL), MCHC = mean cellular hemoglobin content (g/dL), RDW = red blood cell distribution width (%), WBC = number of leukocytes (10^3^/µL), PLT = number of platelets (10^3^/µL).

**Table 1 animals-13-01313-t001:** Signalment of the seven dolphins included in the study. Regarding age, for each animal, the left and right columns represent the age at the first and last samples analyzed, respectively. Age classes were determined according to Venn-Watson and colleagues (2007) [24].

	Animal A		Animal B		Animal C		Animal D		Animal E		Animal F		Animal G	
Age (Years)	21	24	11	14	16	19	14	17	11	14	37	40	6	9
Age (class)	Adult		Adult		Adult		Adult		Adult		Geriatric		Juvenile	
Gender	Male		Female		Male		Male		Female		Female		Female	

**Table 2 animals-13-01313-t002:** Formulas used to calculate the coefficients of variation (CVs), the index of individuality (IoI), and the reference change value (RCV). SD = standard deviation, e = Nepero’s number, VAR is the variance of the data distribution, LN = natural logarithm, x = number of animals, CV_g_ = between-animals coefficient of variation, CV_i_ = within-animal coefficient of variation, CV_a_ = analytical imprecision, Z = 1.96, $\sigma_{i}^{}$ and $\sigma_{a}^{}$ = within-animal and analytical variance of the Gaussian distribution.

	Normal Distribution	Non-Normal Distribution
CV_a,i,g_	$\frac{SD}{mean\;analyte\;value}*100$	$\sqrt{e^{\mathrm{VAR}\;\left(LNx\right)}-1^{*}}$100
IoI	$\frac{CV_{g}}{\sqrt{CV_{i}^{2}\;+\;CV_{a}^{2}}}$	$\frac{CV_{g}}{\sqrt{CV_{i}^{2}\;+\;CV_{a}^{2}}}$
RCV	$Z*\sqrt{2}*\sqrt{CV_{i}^{2}+CV_{a}^{2}}$	$1)+Z*\sqrt{[2*(\sigma_{i}^{2}+\sigma_{a}^{2})]}$
		$2)-Z*\sqrt{[2*(\sigma_{i}^{2}+\sigma_{a}^{2})]}$

**Table 3 animals-13-01313-t003:** Coefficients of variations (CVs), index of individuality (IoI), and reference change value (RCV). CV_g_ = between-animals coefficient of variation, CV_i_ = within-animal coefficient of variation, CV_a_ = analytical imprecision. RBC = number of erythrocytes, Hct = hematocrit, Hgb = hemoglobin, MCV = mean cellular volume, MCHC = mean cellular hemoglobin content, RDW = red blood cell distribution width, WBC = number of leukocytes, PLT = number of platelets.

Analyte	CV_i_ (%)	CV_g_ (%)	CV_a_ (%)	CV_a_/CV_i_	IoI	RCV (%)	
RBC	5.45	7.05	1.75	0.32	1.23	15.86	
Hct	5.86	6.52	2.26	0.39	1.04	17.40	
Hgb	5.79	4.60	1.25	0.22	0.78	16.41	
MCV	3.23	3.38	1.85	0.57	0.90	10.33	
MCHC	3.45	3.37	2.22	0.64	0.82	11.37	
RDW	4.83	6.04	3.20	0.66	1.04	16.06	
WBC	21.69	14.39	2.26	0.16	0.66	53.62	186.51
PLT	13.20	17.34	3.78	0.29	1.26	30.05	

**Table 4 animals-13-01313-t004:** Comparison of within-animal coefficient of variation (CV_i_) for hematological measurands in different species. RBC = number of erythrocytes, Hct = hematocrit, Hgb = hemoglobin, MCV = mean cellular volume, MCHC = mean cellular hemoglobin content, RDW = red blood cell distribution width, WBC = number of leukocytes, PLT = number of platelets, n.a. = not assessed.

CV_i_	RBC (10^9^/µL)	Hct (%)	Hgb (g/dL)	MCV (fL)	MCHC (g/dL)	RDW (%)	WBC (10^3^/µL)	PLT (10^3^/µL)
Bottlenose dolphins	5.45	5.86	5.79	3.23	3.45	4.83	21.69	13.20
Cows [18]	5.40	6.10	5.30	2.30	1.90	n.a.	9.10	11.20
Elephants [19]	7.20	7.40	8.0	0.80	1.70	1.10	9.70	17.40
Horses [17]	6.29	6.18	6.30	0.65	0.70	1.62	8.37	16.54
Dogs [13]	6.0	6.20	6.10	2.10	2.60	2.60	19.60	14.0
Dogs [14]	7.80	7.69	7.76	1.60	2.93	4.52	18.98	27.07
Humans [26]	3.20	2.70	2.85	1.40	1.06	2.70	11.40	9.10

## Data Availability

Raw data are available by writing an email to the corresponding author (federico.bonsembiante@unipd.it).

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
