# Peer review of "Biological Variation and Reference Change Value of Routine Hematology Measurands in a Population of Managed Bottlenose Dolphins (Tursiops truncatus)"

_animals, 2023, doi:10.3390/ani13081313_

Round 1

Reviewer 1 Report

The paper shows up once again, that intra-individual laboratory results are useful to overcome the lack of population based reference limits in rare species. This is not new, but has not been shown for dolphins before.

Author Response

Thanks to the reviewer for his/her comment. We are glab about his/her comment.

Reviewer 2 Report

Laboratory data is usually interpreted with respect to population-based reference intervals but for some analytes in which intraindividual (within animal) variation is less than interindividual (between animal) variation, subject-based reference values may be more applicable.If the biologic (day-to-day) variation within an individual animal is larger than that seen between animals in the population, population-based intervals are appropriate (see figure to the right). If the variation within an individual is less than that in the population, this means that population-based intervals may not be the best method of detecting an abnormal test result in that individual (see figure to the right). In the latter scenario, a change in the individual animal’s result from an established baseline for that animal may be the best way of picking up a test abnormality that is due to disease. This change, called the reference change value (also called the critical difference), gives rise to the subject-based reference value. So if the test result changes from baseline more than the reference change value, this supports a true change versus that just due to chance. However, as for any test result, the change in analyte values may be due to pre-analytical variables (e.g. breed) versus disease (so you still have to figure this out). 

The foregoing indicates that there is a great need to determine intraindividual variability in as many animal species as possible. Such a longitudinal and intraindividual approach is a step toward personalized veterinary medicine, which represents the future of laboratory diagnostics.

The work is written well and of high quality, and methods are applied according to recommended laboratory standards. The authors obtained useful results. The authors also mentioned certain limitations that they had. Let's say they only used two samples, although ideally a larger number would be used. However, two samples are acceptable and were used in a recent study by Nikolić et al (2022): The effects of biological and health characteristics of dogs on intraindividual variability of blood parameters, doi: 10.55730/1300-0128.4208. This work can explain the methodology they used.

I would suggest that the authors improve the work in several ways:

1) Graphically present the obtained subject vs. population based reference interval, by specifying all known ranges of reference values for dolphins and within them the mean and variance for each dolphin separately in this trial. Let the results presented at the following link be your guide: https://eclinpath.com/test-basics/reference-intervals/print-18/. Here you will get 8 new graphs, for each parameter separately.

2) CVi is a relative value, so the CVi of different animal species can be compared with each other. It would be good if you could present for each of the examined parameters the relationship between the obtained CVi in dolphins and the CVi of other animal species, as well as humans. This will significantly contribute to a better understanding of the importance and variability in dolphins compared to other populations. Let the manuscript you have already cited (Kovačević et al) be your guide for that, in which there are conceptual graphs and a description of the method by which the data were obtained. Here you will also get 8 new charts.

By improving the presented results and expanding the discussion and other elements in order to support those results, we will get a high-quality and communicative manuscript.

This manuscript should be accepted for publication, because such results obtained on dolphins are truly a rarity and represent great value for laboratory work.

Author Response

Thanks to the reviewer to his/her comments that improve the quality of the article. For his/her convenience here there are his/her comments (in black) and the answers to the comments (in red).

Laboratory data is usually interpreted with respect to population-based reference intervals but for some analytes in which intraindividual (within animal) variation is less than interindividual (between animal) variation, subject-based reference values may be more applicable.If the biologic (day-to-day) variation within an individual animal is larger than that seen between animals in the population, population-based intervals are appropriate (see figure to the right). If the variation within an individual is less than that in the population, this means that population-based intervals may not be the best method of detecting an abnormal test result in that individual (see figure to the right). In the latter scenario, a change in the individual animal’s result from an established baseline for that animal may be the best way of picking up a test abnormality that is due to disease. This change, called the reference change value (also called the critical difference), gives rise to the subject-based reference value. So if the test result changes from baseline more than the reference change value, this supports a true change versus that just due to chance. However, as for any test result, the change in analyte values may be due to pre-analytical variables (e.g. breed) versus disease (so you still have to figure this out). 

Thanks to the reviewer. We discussed the impact of pre-analytical errors at lines 260-267 of the revised manuscript.

The foregoing indicates that there is a great need to determine intraindividual variability in as many animal species as possible. Such a longitudinal and intraindividual approach is a step toward personalized veterinary medicine, which represents the future of laboratory diagnostics.

The work is written well and of high quality, and methods are applied according to recommended laboratory standards. The authors obtained useful results. The authors also mentioned certain limitations that they had. Let's say they only used two samples, although ideally a larger number would be used. However, two samples are acceptable and were used in a recent study by Nikolić et al (2022): The effects of biological and health characteristics of dogs on intraindividual variability of blood parameters, doi: 10.55730/1300-0128.4208. This work can explain the methodology they used.

In our study, we included six samples for each animal. We did not have the possibility to include 10-15 animals as proposed by Freeman and colleagues because our population was made up of seven dolphins. We better specify the number of CBC exams for each animal at lines 102-103 of the revised manuscript.

I would suggest that the authors improve the work in several ways:

  • Graphically present the obtained subject vs. population based reference interval, by specifying all known ranges of reference values for dolphins and within them the mean and variance for each dolphin separately in this trial. Let the results presented at the following link be your guide: https://eclinpath.com/test-basics/reference-intervals/print-18/. Here you will get 8 new graphs, for each parameter separately.

Thanks to the reviewer. We added the suggested graphs in the text (Figure 1). We compared our results with the pRIs published in the article of Lauderdale and colleagues (2021) and in the 3rd edition of the CRC Handbook of Marine Mammals Medicine.

  • CVi is a relative value, so the CVi of different animal species can be compared with each other. It would be good if you could present for each of the examined parameters the relationship between the obtained CVi in dolphins and the CVi of other animal species, as well as humans. This will significantly contribute to a better understanding of the importance and variability in dolphins compared to other populations. Let the manuscript you have already cited (Kovačević et al) be your guide for that, in which there are conceptual graphs and a description of the method by which the data were obtained. Here you will also get 8 new charts.

Thanks to the reviewer. We added a table in which the CVi of dolphins and other mammals species were compared (Table 4).

By improving the presented results and expanding the discussion and other elements in order to support those results, we will get a high-quality and communicative manuscript.

This manuscript should be accepted for publication, because such results obtained on dolphins are truly a rarity and represent great value for laboratory work.

Reviewer 3 Report

I have carefully read the manuscript of Bonsembiante and colleagues focused of the biological variations of routine hematological parameters in dolphins. It is a very interesting study that explore an important topic that deserves to be further investigated, in dolphins as well as in other exotic and non-conventional species. However, the manuscript needs to be partially improved to clarify some aspects and better put in light the usefulness of the study. The points that should be amended don’t concern the quality of the study, but the lack of some important information in the text.

First, the study design should be better presented, with a clear list of the number of samples/year/individual that should be provided, maybe using a table to resume them. In the text, it is stated that the total number of values was 504, but it is not possible to find out how this count was made. Furthermore, I also suggest to briefly explain if the age can influence these biological values as in other species, since the studies population belongs to different ages.

Second, I think that the authors should also present, and discuss, the data about each parameter (e.g. mean, median, min-max…). It is clear that the main aim of the authors is to evaluate the value variations and their impact of hematological analysis reliability, but this could help the reader to better understand these main results and to see the direct implication in clinical application. Moreover, the report of these biological values can increase the article interest since it would add some references in literature of a species less studied than others. These data should be also discussed and compared to other from bibliography, to increase the knowledge about the reference intervals in dolphins and the clinical perspectives.

Third, it should be interesting if the authors dedicate some sentences to the automate they used to obtain the hematological parameters and in particular how they asses the reliability of this automate. Also, they should also explain why the WBC count could not be done with the same automate. For both, it could be interesting to cite articles that confirm the approach that the authors have chosen. As for all the exotic and less studied species, it is very useful to provide more information as possible (about the technical part as well as the part concerning dolphin hematology) to increase the debate and the knowledge.    

Author Response

Thanks to the reviewer for his/her comments that improve the quality of the article. For his/her convenience here there are the reviewer's comments (in black) and the answers (in red).

I have carefully read the manuscript of Bonsembiante and colleagues focused of the biological variations of routine hematological parameters in dolphins. It is a very interesting study that explore an important topic that deserves to be further investigated, in dolphins as well as in other exotic and non-conventional species. However, the manuscript needs to be partially improved to clarify some aspects and better put in light the usefulness of the study. The points that should be amended don’t concern the quality of the study, but the lack of some important information in the text.

First, the study design should be better presented, with a clear list of the number of samples/year/individual that should be provided, maybe using a table to resume them. In the text, it is stated that the total number of values was 504, but it is not possible to find out how this count was made. Furthermore, I also suggest to briefly explain if the age can influence these biological values as in other species, since the studies population belongs to different ages.

Thanks to the reviewer. We added the information requested on lines 102-103; 162-164; 265-269 and we added a line in Table 1 with the age class for each animal.

Second, I think that the authors should also present, and discuss, the data about each parameter (e.g. mean, median, min-max…). It is clear that the main aim of the authors is to evaluate the value variations and their impact of hematological analysis reliability, but this could help the reader to better understand these main results and to see the direct implication in clinical application. Moreover, the report of these biological values can increase the article interest since it would add some references in literature of a species less studied than others. These data should be also discussed and compared to other from bibliography, to increase the knowledge about the reference intervals in dolphins and the clinical perspectives.

Thanks to the reviewer. According to reviewer 2 we added graphs that showed the comparison of individual reference intervals to the population-based reference intervals present in the literature (Figure 1).

Third, it should be interesting if the authors dedicate some sentences to the automate they used to obtain the hematological parameters and in particular how they asses the reliability of this automate. Also, they should also explain why the WBC count could not be done with the same automate. For both, it could be interesting to cite articles that confirm the approach that the authors have chosen. As for all the exotic and less studied species, it is very useful to provide more information as possible (about the technical part as well as the part concerning dolphin hematology) to increase the debate and the knowledge. 

Thanks to the reviewer. We added this information at lines 197-203 and 220-224.   

Round 2

Reviewer 3 Report

I want to thank you the authors that made prompt and efficient efforts to definitively improve this interesting manuscript, which is now fully suitable for the publication in the journal.